# Effect of Pinoresinol and Vanillic Acid Isolated from *Catalpa bignonioides* on Mouse Myoblast Proliferation via the Akt/mTOR Signaling Pathway

**DOI:** 10.3390/molecules27175397

**Published:** 2022-08-24

**Authors:** Seo-Young Kim, Sung-Pil Kwon, SeonJu Park, Su-Hyeon Cho, Youngse Oh, Seung Hyun Kim, Yoon Ho Park, Hyun Suk Jung, Deug-chan Lee, Hoibin Jeong, Kil-Nam Kim

**Affiliations:** 1Chuncheon Center, Korea Basic Science Institute (KBSI), Chuncheon 24341, Korea; 2Honam National Institute of Biological Resources, Mokpo 58762, Korea; 3Research & Development Center, Chungdam CDC JNPharm LLC., Chuncheon 24341, Korea; 4Department of Biomedical Technology, Kangwon National University, Chuncheon 24341, Korea; 5Yonsei Institute of Pharmaceutical Sciences, College of Pharmacy, Yonsei University, Incheon 21983, Korea; 6Department of Biochemistry, College of Natural Sciences, Kangwon National University, Chuncheon 24341, Korea; 7Korea Research Institute of Bioscience and Biotechnology (KRIBB), Daejeon 24341, Korea

**Keywords:** *Catalpa bignonioides*, pinoresinol, vanillic acid, IGF-1 signaling

## Abstract

Growth and maintenance of skeletal muscle is essential for athletic performance and a healthy life. Stimulating the proliferation and differentiation of muscle cells may help prevent loss of muscle mass. To discover effective natural substances enabling to mitigate muscle loss without side effects, we evaluated muscle growth with several compounds extracted from *Catalpa bignonioides* Walt. Among these compounds, pinoresinol and vanillic acid increased C2C12, a mouse myoblast cell line, proliferation being the most without cytotoxicity. These substances activated the Akt/mammalian target of the rapamycin (mTOR) pathway, which positively regulates the proliferation of muscle cells. In addition, the results of in silico molecular docking study showed that they may bind to the active site of insulin-like growth factor 1 receptor (IGF-1R), which is an upstream of the Akt/mTOR pathway, indicating that both pinoresinol and vanillic acid stimulate myoblast proliferation through direct interaction with IGF-1R. These results suggest that pinoresinol and vanillic acid may be a natural supplement to improve the proliferation of skeletal muscle via IGF-1R/Akt/mTOR signaling and thus strengthen muscles.

## 1. Introduction

There is a gradual loss of skeletal muscle mass and strength associated with age, known as sarcopenia. This is inevitable in most people and is considered a major cause of disability and fragility in older adults, reducing their quality of life. It begins at the age of 40 years, with a decrease in muscle mass of up to 8% decennially; this rate may double by the age of 70 years [1]. For a healthy life of old age, many researchers have been interested in finding edible substances that maintain and stimulate skeletal muscle mass. Skeletal muscle growth is achieved through proliferation and differentiation of muscle fibers, and the insulin-like growth factor 1 (IGF-1)/Akt/mammalian target of rapamycin (mTOR) signaling pathway, which stimulates them, is considered to be a master regulator of skeletal myogenesis [2,3]. Although many results have been proposed that IGF-1 upregulates muscle weight and mitigates atrophy in animal studies, there are less dramatic impacts in humans [4,5], encouraging further studies on other compounds that can achieve the desired effect. Therefore, compounds that stimulate skeletal myogenesis and IGF-1 can help overcome muscle atrophy and facilitate muscle growth.

Steroidal androgens of the protein anabolic steroid class are drugs typically used to stimulate muscle enhancement. The biological efficacy of these drugs is demonstrated by muscle mass increase, growth spurts in premature children, and bone loss attenuation in older adults [6,7], These compounds are prescribed for various therapeutic purposes. However, long-term or excessive usage may cause side effects, such as skin diseases [7] or reproductive and endocrine functional deterioration [8,9]. Aiming to treat impaired myogenesis via metabolic changes, Belli et al. reported that trimetazidine, a metabolic modulator drug, may induce myoblast differentiation in a cell line and increase muscle strength in mice [10]. Although trimetazidine is used in the treatment of angina, its long-term use has been associated with gastrointestinal disturbances, vomiting, and nausea [11]. Therefore, there is a need for a compound that prevents skeletal muscle mass loss without increasing side effect risks.

*Catalpa bignonioides* Walt. (Bignoniaceae) is a bean tree native to southeastern America. It has been used in traditional medicinal practices for respiratory diseases, scrofulous ulcers, and helminthic infections, among others. A previous study on the bioactivity of *C. bignonioides* extracts [12] has shown that its flowers, leaves, and capsule valves have antioxidant activity [13]. Other studies revealed that catalpic acid, a conjugated triene fatty acid abundant in *C. bignonioides*, may improve insulin homeostasis by decreasing fat accumulation in the adipose tissue of mice [14]. Recently, we identified the constituents of the methanol extract of *C. bignonioides* fruits through phytochemical analysis, revealing that some compounds had properties stimulating α-glucosidase inhibition and insulin secretion [15]. Although studies evaluating the biological activities of compounds isolated from *C. bignonioides* are ongoing, the impact of *C. bignonioid*-derived substances on muscle loss remains unclear. Herein, we aimed to identify components from the fruits of *C. bignonioides* extracts that promote muscle proliferation and differentiation via Akt/mTOR signaling.

## 2. Materials and Methods

### 2.1. Materials

Dulbecco’s Modified Eagle Medium (DMEM) was purchased from Welgene (Gyeongsangbuk-do, Korea). Fetal bovine serum (FBS) was purchased from Omega Scientific, Inc. (Tarzana, CA, USA). Penicillin and streptomycin were purchased from Invitrogen (Carlsbad, CA, USA). Horse serum (HS), 3-(4,5-dimethylthiazol-2-yl)-2,5-diphenyltetrazolium bromide (MTT), dimethyl sulfoxide (DMSO), and radioimmunoprecipitation assay buffer were purchased from Sigma-Aldrich (St. Louis, MO, USA). NuPAGE 4–12% Bis-Tris gel was purchased from Life Technologies (Carlsbad, CA, USA). Ployvinylidine fluoride membranes were purchased from Bio-Rad Laboratories (Hercules, CA, USA). Rabbit anti-mouse phospho-Smad2(Ser465/467)/3(Ser423/425) (Cat#8828), rabbit anti-mouse Smad4 (Cat#46535), rabbit anti-mouse phospho-Akt (Ser473, Cat#4060), rabbit anti-mouse phospho-mTOR (Ser2448, Cat#5536), rabbit anti-mouse phospho-ribosomal protein S6 kinase (p70S6K) (Thr421/Ser424, Cat#9204), rabbit anti-mouse phospho-eukaryotic initiation factor 4E-binding protein 1 (4E-BP1) (Thr37/46, Cat#2855), goat anti-rabbit IgG (Cat#7074), and goat anti-mouse IgG (Cat#7076) were purchased from Cell Signaling Technology (Danvers, MA, USA). Mouse anti-mouse myoblast determination protein 1 (MyoD) (Cat#sc-377460), mouse anti-mouse myogenin (Cat#sc-12732), and mouse anti-mouse β-actin (Cat#sc-47778) were purchased from Santa Cruz Biotechnology (Dallars, TX, USA).

### 2.2. Plant Material

*C. bignonioides* fruits were collected from the Medicinal Herb Garden, College of Pharmacy, Seoul National University in Goyang, Gyeonggi-do, Korea in 2021 and authenticated by Sang Il Han, a general manager of the Medicinal Herb Garden. A voucher specimen (CB202106) was deposited at the Korea Basic Science Institute (Chuncheon, Korea).

### 2.3. Extraction and Isolation

The dried fruits of *C. bignonioides* (1.3 kg) were extracted with MeOH (5 L × 3 times) under sonication at 30 °C for 4 h to yield an extract (91.0 g), which was then dissolved in H_2_O and successively partitioned using CHCl_3_ and EtOAc to obtain CHCl_3_ (CB1, 16.0 g), EtOAc (CB2, 2.5 g), and H_2_O (CB3, 71.0 g) extracts after removing the solvents in vacuo.

The CHCl_3_ fraction was subjected to silica gel CC and eluted with a gradient of hexane:acetone (40:1 → 2.5:1, *v*/*v*) and CHCl_3_:MeOH (20:1 → 2.5:1, *v*/*v*) to yield nine subfractions, CB1A (3.0 g), CB1B (2.4 g), CB1C (1.0 g), CB1D (1.5 g), CB1E (1.0 g), CB1F (1.2 g), CB1G (0.8 g), CB1H (1.0 g), and CB1I (0.5 g). The CB1F fraction was applied to a YMC RP-18 column, which was eluted with MeOH:H_2_O (1.3:1, *v*/*v*), yielding four smaller fractions, CB1F1 (58.2 mg), CB1F2 (41.5 mg), CB1F3 (18.4 mg), and CB1F4 (16.5 mg). The CB1F1 fraction was subjected to HPLC using a J’sphere ODS H-80 250 mm × 20 mm column, eluted with 28% MeCN in H_2_O at a flow rate of 3 mL/min to yield **1** (6.8 mg) and **2** (7.1 mg). The CB1F2 fraction was subjected to the same HPLC conditions, except that the elution solvent was 40% MeCN in H_2_O, to afford **3** (8.1 mg).

The H_2_O fraction (CB3, 71.0 g) was chromatographed on a Diaion HP-20 column and eluted with H_2_O containing increasing concentrations of MeOH (25, 50, and 100%) to obtain three subfractions, CB3A (10.0 g), CB3B (13.0 g), and CB3C (6.0 g). The CB3B fraction was subjected to silica gel CC and eluted with a gradient of CHCl_3_:MeOH (10:1 → 2.5:1, *v*/*v*) to yield three subfractions, CB3B1 (2.0 g), CB3B2 (2.7 g), and CB3B3 (2.0 g). The CB3B1 fraction was applied to a silica gel column and eluted with CHCl_3_:MeOH:H_2_O (5:1:0.1, *v*/*v*), CB3B11 (31.0 mg), CB3B12 (96.0 mg), CB3B13 (82.0 mg), CB3B14 (150.4 mg), CB3B15 (66.7 mg), and CB3B16 (213.8 mg). The CB3B14 fraction was subjected to HPLC purification under 40% MeCN to yield **13** (26.2 mg) and **14** (6.8 mg). The CB3B16 fraction was subjected to the same HPLC conditions, except that elution with 23% MeCN in H_2_O afforded **15** (140.0 mg). The CB3C fraction was subjected to silica gel CC and eluted with a gradient of CHCl_3_:MeOH (10:1 → 2.5:1, *v*/*v*) to yield three subfractions, CB3C1 (0.4 g), CB3C2 (1.5 g), and CB3C3 (1.0 g). The CB3C1 fraction was applied to a YMC RP-18 column and eluted with MeOH:H_2_O (1:1, *v*/*v*), yielding three smaller fractions, CB3C11 (0.2 g), CB3C12 (55.5 mg), and CB3C13 (14.0 mg). The CB3C11 fraction was subjected to HPLC using a J’sphere ODS H-80 250 mm × 20 mm column, eluted with MeCN:H_2_O (18:82), and a flow rate of 3 mL/min to yield **4** (55.9 mg), **5** (7.3 mg), and **6** (14.3 mg). The CB3C2 fraction was applied to a YMC RP-18 column, which, when eluted with MeOH:H_2_O (1:1, *v*/*v*), yielded three smaller fractions, CB3C21 (0.2 g), CB3C22 (0.6 g), and CB3C23 (0.2 g). The CB3C21 fraction was subjected to HPLC using a J’sphere ODS H-80 250 mm × 20 mm column, eluted with MeCN:H_2_O (30:70), and a flow rate of 3 mL/min to yield **7** (35.5 mg), whereas the CB3C23 fraction gave **8** (30.1 mg), **9** (18.5 mg), and **10** (6.3 mg). The CB3C3 fraction was applied to a YMC RP-18 column, which when eluted with MeOH:H_2_O (1.4:1, *v*/*v*), yielded four smaller fractions, CB3C31 (42.8 mg), CB3C32 (0.1 g), CB3C33 (30.8 mg), and CB3C34 (18.6 g). The CB3C32 fraction was subjected to HPLC using a J’sphere ODS H-80 250 mm × 20 mm column, eluted with MeCN:H_2_O (25:75), at a flow rate of 3 mL/min to yield **11** (11.3 mg). The CB3C34 fraction was subjected to the same HPLC conditions, except that the eluding solvent was MeCN:H_2_O (23:77), to afford **12** (7.1 mg).

### 2.4. Cell Culture and Differentiation

C2C12 cells (mouse myoblast cell line) were maintained in DMEM supplemented with 10% FBS, penicillin (100 U/mL), and streptomycin (100 μg/mL). For differentiation, when the cell confluence reached approximately 80%, the medium was replaced with DMEM containing 2% HS. DMEM containing 2% FBS was replaced every other day, and differentiation proceeded for 6 days. All cell cultures were maintained at 37 °C in a 5% CO_2_ incubator.

### 2.5. Cytotoxicity Assay

The cytotoxicity of compounds extracted from *C. bignonioides* was assessed using a colorimetric assay. In total, 1 × 10^5^ cells/mL were seeded in 96-well plates and incubated with the test compounds for 24 h. Thereafter, 100 μg/mL of MTT were added to each well. After 2.5 h incubation at 37 °C, the supernatants were aspirated, and cells were treated with DMSO to dissolve the formazan crystals. The absorbance of the colored solution was determined at 540 nm using a SpectraMax M2/M2e spectrophotometer (Molecular Devices, San Jose, CA, USA).

### 2.6. Cell Proliferation Activity

To measure skeletal muscle cell proliferation activity, C2C12 cells were seeded at a concentration of 5 × 10^4^ cells/mL in a 96-well plate, and the medium was replaced with DMEM containing 2% HS two days later to induce differentiation. The compounds were added whenever the medium (DMEM with 2% HS) was changed every other day. Six days after differentiation induction, cell proliferation was assessed using a 5-bromo-2′-deoxyuridine (BrdU) assay kit (Millipore, Billerica, MA, USA).

### 2.7. Western Blot

Cell lysates were prepared using radioimmunoprecipitation assay buffer. Quantified protein lysates were loaded onto NuPAGE 4–12% Bis-Tris gels, which were then blotted onto a polyvinylidene fluoride membrane. Primary antibodies, including rabbit anti-mouse phospho-Smad2/3, rabbit anti-mouse Smad4, rabbit anti-mouse phospho-Akt, rabbit anti-mouse phospho-mTOR, rabbit anti-mouse phospho-p70S6K, rabbit anti-mouse phospho-4E-BP1, mouse anti-mouse MyoD, mouse anti-mouse myogenin, and mouse anti-mouse β-actin antibodies were diluted at 1:1000 and incubated overnight at 4 °C. Secondary antibodies, including goat anti-rabbit IgG and goat anti-mouse IgG, were diluted at 1:3000 and incubated for 1.5 h at 25 °C. Signals were developed using the SuperSignal West Femto Trial Kit (Thermo Fisher Scientific; Waltham, MA, USA), and images were acquired using Fusion FX (Vilber Lourmat Ste, Collegien, France) or VISQUE^Ⓡ^ InVivo Smart-LF (Vieworks. Co., Ltd., Anyang-si, Korea).

### 2.8. In Silico Molecular Docking Simulation

In order to verify the potential active chemicals that are able to act as IGF-1 receptor (IGF-1R) agonists, a molecular docking study was performed. First, the crystal structure of IGF-1R (PDB ID: 1IGR) was obtained from the Protein Data Bank (PDB, http://www.pdb.org; accessed on 1 January 2022). Hereafter, a docking simulation was performed using AUTODOCK VINA to investigate if the potential active chemicals bind to IGF-1R [16] and analyzed for IGF-1R and chemicals interaction by LIGPLOT [17]. The 2-dimensional interaction map shows the hydrogen bond in green and labeled non-ligand residues involved in hydrophobic contact in red. In this research, we used 5 ligands, wherein the dihydrotestosterone (DHT) docked with IGF-1R simulation result was considered as a positive control, while dieckol and 2,7′-phloroglucinol-6,6′-bieckol docking results were correlated with a previously published paper [18].

### 2.9. Statistical Analysis

Variables were compared using two-tailed one-way ANOVA with Tukey’s post hoc test using Prism software (Version 4.00; GraphPad Inc.; La Jolla, CA, USA). The findings were considered statistically significant at *p*-values of <0.05.

## 3. Results

### 3.1. Pinoresinol and Vanillic Acid Facilitated C2C12 Cell Proliferation

To examine the effect of 15 constituents from *C. bignonioides* fruit extracts on skeletal muscle growth, we measured cell proliferation activity in cell line C2C12 (Figure 1) [15]. 

Prior to examining the effect on cell proliferation, the viability of C2C12 myoblasts was evaluated using MTT colorimetric assay. We confirmed that none of the compounds, except compound **9** (6-*O*-*trans*-feruloyl catalpol), showed toxicity to C2C12 cells at a concentration of 50 μM (Figure 2A).

Next, the proliferation effect of skeletal muscle cells was measured during the differentiation period by BrdU cell proliferation assay [19,20,21,22], except for compound **9**, which was excluded because of its toxicity. Compared to the control conditions, most compounds triggered approximately a 1.2-fold increase in cell proliferation (Figure 2B). Among them, compound **3** (pinoresinol) and compound **5** (vanillic acid) showed a 1.8-fold increase.

To determine effects at low concentrations, cell proliferation was measured in a dose-dependent manner using pinoresinol and vanillic acid, which had the greatest effect among the compounds. Pinoresinol and vanillic acid showed cell proliferation activity during the differentiation phase even at concentrations of 6.25 μM and 12.5 μM, respectively (Figure 3).

### 3.2. Pinoresinol and Vanillic Acid Stimulated Akt/Mtor Signaling Pathway in C2C12 Cells

Myogenesis, differentiation, and maturation of skeletal muscle cells are regulated by signaling pathways activated by the transforming growth factor beta (TGF-β) superfamily [23]. To determine whether cell proliferation activity of pinoresinol and vanillic acid was mediated by TGF-β signaling, we performed Western blotting for Smad proteins in C2C12 differentiated cells treated with either compound. We found that p-Smad2 and p-Smad3 were not significantly downregulated by pinoresinol or vanillic acids (Figure 4). Furthermore, the expression level of Smad4 was mildly decreased by these compounds.

IGF-1 signaling is a positive regulator of muscle cell proliferation and differentiation [24]. To examine the activation of IGF-1 signaling by pinoresinol and vanillic acid, we analyzed the phosphorylation levels of Akt, mTOR, and p70S6K using Western blotting. We found that p-Akt, p-mTOR, and p-p70S6K levels were increased by these compounds (Figure 4). In addition, these compounds decreased the phosphorylation of 4E-BP1, indicating suppression of its growth inhibitory function. Furthermore, essential regulators that induce muscle differentiation, such as MyoD and myogenin, were significantly increased in C2C12 myoblasts treated with pinoresinol or vanillic acid. These results suggest that pinoresinol and vanillic acid stimulate myogenic differentiation by regulating Akt/mTOR signaling in mouse muscle cells.

### 3.3. Pinoresinol and Vanillic Acid Were Docked into Igf-1 Receptor through in Silico Analysis

We showed that the levels of protein expression involved in IGF-1 signaling were highly upregulated in pinoresinol- or vanillic-acid-treated C2C12 cells. To determine the potential of these two active compounds to bind to the IGF-1R, we simulated the docking of pinoresinol, vanillic acid, and IGF-1R. Docking of the ligand–protein complexes was successful, as both compounds stably posed to the active sites of IGF-1R (Figure 5A,B). In addition, DHT, used as a positive control [18,25,26,27], was shown in the docking simulation to stably pose the active site of IGF-1R with a similar low binding energy value as pinoresinol or vanillic acid (Figure 5C and Table 1).

## 4. Discussion

In this study, we demonstrated the effects on myoblast proliferation of two compounds extracted from *C. bignonioides*: pinoresinol and vanillic acid. Pinoresinol is a biologically active ligand mainly found in medicinal plants, such as *Styrax* sp. and *Forsythia suspense*, and in olive oil [28,29]. Pinoresinol possesses antioxidant, anti-inflammatory, and antifungal activities, and has been used in traditional medicine for a long time [28,29,30]. A recent study demonstrated that the effect of defatted sesame seeds in alleviating hypoglycemia is mediated by the inhibitory function of α-glucosidase activity of pinoresinol [31]. In a study on anticancer activity, pinoresinol was found to induce apoptosis and to suppress migration in human liver cancer cells [32]. Vanillic acid, an oxidative form of vanillin and phenolic compound, has also been used in folk medicine. Vanillic acid is abundant in the root of *Angelica sinensis*, also known as female ginseng, and exhibits antioxidant, anticancer, and cardioprotective activities [33,34,35]. However, no studies have examined the effect of vanillic acid or pinoresinol on muscle cells or muscle-related diseases. This study is the first to show that pinoresinol and vanillic acid extracted from an edible plant promote proliferation in myoblasts.

Skeletal muscles play pivotal roles in physical activity and energy metabolism. Growth and maintenance of skeletal muscle are essential for the management of natural aging of muscles, in which skeletal muscle decreases. The growth and differentiation of skeletal muscle are regulated by negative and positive regulators, specifically, TGF-β and IGF-1/Akt/mTOR pathways, respectively. TGF-β family members, including myostatin, are known to inhibit myogenic differentiation in cultured primary myoblasts or myoblast cell lines, such as C2C12 [36,37,38]. C2C12 myoblasts lacking the TGF-β1 signal due to mutation of TβR II, a component of the TGF-β receptor, cannot form myotubes [39]. IGF-1 binds to IGF-1R, phosphorylates insulin receptor substrate-1 (IRS-1), an adaptor protein inside the cell, and sequentially phosphorylates phosphoinositide 3-kinase (PI3K) and Akt. Meanwhile, mTOR, which is a downstream target of Akt, phosphorylates p70S6K to promote protein synthesis and 4E-BP1 to induce translation initiation [40]. The Akt/mTOR/p706K pathway mediates downstream signaling of IGF-1 to promote protein synthesis and body growth and promotes both cell proliferation and differentiation in cultured myoblasts [41]. IGF-1 also induces cell differentiation by inducing the expression of myogenic regulatory factors (MEFs) such as MyoD and myogenin during myogenic differentiation [42]. In the L6E9 cell line (rat-derived myoblast) in the process of differentiation, when IGF-1 was overexpressed, myogenin levels increased, and myotubes became enlarged [43]. We demonstrated that natural substances with muscle proliferation activity, pinoresinol and vanillic acid increased the expression of MyoD and myogenin, and phosphorylation of downstream targets of IGF-1, including Akt, mTOR, and p70S6K. Furthermore, through in silico docking analysis, we showed the potential of these substances to bind to the active site of IGF-1R. 

Although TGF-β is known to negatively regulate myoblast differentiation by suppressing the expression of two MRFs (MyoD and myogenin) through Smad3 [23,44,45,46], our study confirmed that the myogenic effects of pinoresinol and vanillic acid were unlikely achieved through the TGF-β/Smad pathway.

The proliferative effect of both compounds was evaluated during the differentiation process of C2C12; it is known that cell cycle is arrested. However, since the function of skeletal muscle is represented by differentiation proceeding with an increase in the number of muscle cells, we conducted measuring of cell proliferation in differentiated muscle cells as well that established in previously published papers [18,47,48].

Pinoresinol and vanillic acid, demonstrated in this study to promote the proliferation and differentiation of myoblast cells, could be applied to alleviating human diseases such as muscle loss. There have been studies confirming biological effects using those compounds in vivo. The injection of pinoresinol into rats with ischemic damage restored blood flow distribution and reduced the level of reactive oxygen species, whose accumulation is known to be associated with muscle atrophy and muscle weakness [49,50]. Vanillic acid was reported as a therapeutic option for patients lacking coenzyme Q_10_, and the lack of coenzyme Q_10_ is highly correlated with the risk of age-related muscle loss [51,52]. Although further studies should be done, it is expected that pinoresinol and vanillic acid may be effective in improving muscle strength in humans as well.

## 5. Conclusions

In summary, this study demonstrated that pinoresinol and vanillic acid isolated from *C. bignonioides* stimulated the Akt/mTOR pathway to promote proliferation and differentiation of myoblasts, suggesting that they bind to IGF-1R, upstream of Akt/mTOR. These findings may provide fundamental data for pinoresinol and vanillic acid as natural muscle building supplements to attenuate muscle loss.

## Figures and Tables

**Figure 1 molecules-27-05397-f001:**
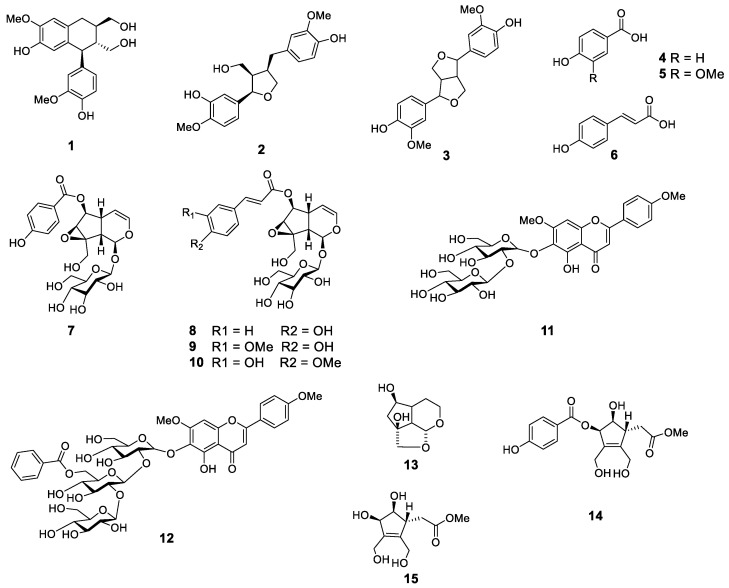
Chemical structures of compounds **1**–**15**. **1**, isolariciresinol; **2**, (+)-lariciresinol; **3**, pinoresinol; **4**, 4-hydroxybenzoic acid; **5**, vanillic acid; **6**, *trans*-*p*-coumaric acid; **7**, catalposide; **8**, specioside; **9**, 6-*O*-*trans*-feruloyl catalpol; **10**, minecoside; **11**, 5,6-dihydroxy-7,4′-dimethoxyflavone-6-*O*-sophoroside; **12**, 5,6-dihydroxy-7,4′-dimethoxyflavone-6-*O*-[6′′′-benzoyl-β-D-glucopyranosyl-(1→2)-β-D-glucopyranosyl-(1→2)]-β-D-glucopyranoside; **13**, des-*p*-hydroxybenzoyl-3-deoxycatalpin; **14**, (7*R*)-3-methoxy-(7-*O*-*p*-hydroxybenzoyl)eucommic acid; **15**, (7*R*)-3-methoxy-hydroxyeucommic acid.

**Figure 2 molecules-27-05397-f002:**
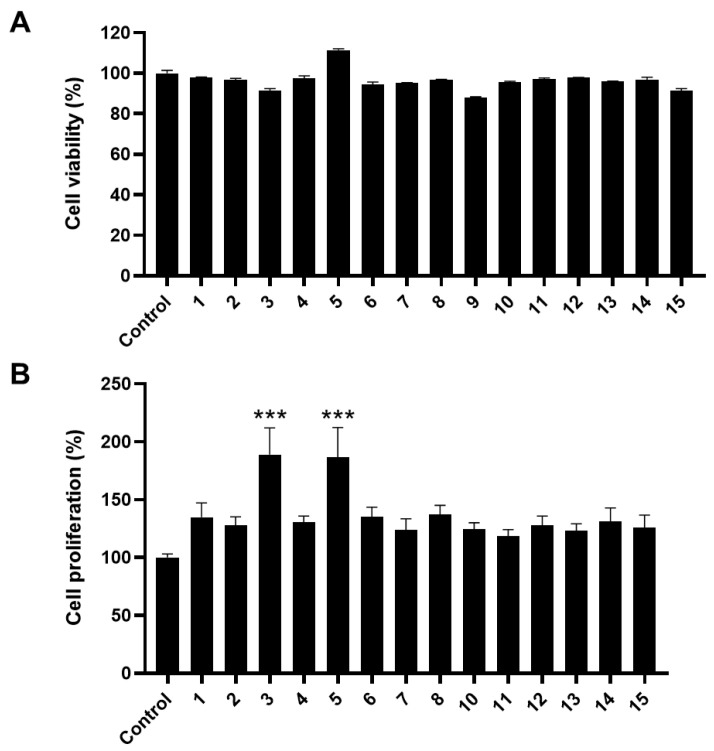
Cell proliferative effect of components from *C. bignonioides* fruits on mouse myoblasts. (**A**) Cytotoxicity measured by MTT colorimetric assay for 50 μM concentration of compound **1**–**15** isolated from *C. bignonioides* on C2C12 cells. (**B**) Cell proliferation activity measured by BrdU assay of compound **1**–**8** and **10**–**15** at 50 μM concentration on C2C12 cells during the differentiation period. Data are reported as mean ± SEM, obtained from at least triplicate determinations. *** indicates *p* < 0.001 calculated by one-way ANOVA.

**Figure 3 molecules-27-05397-f003:**
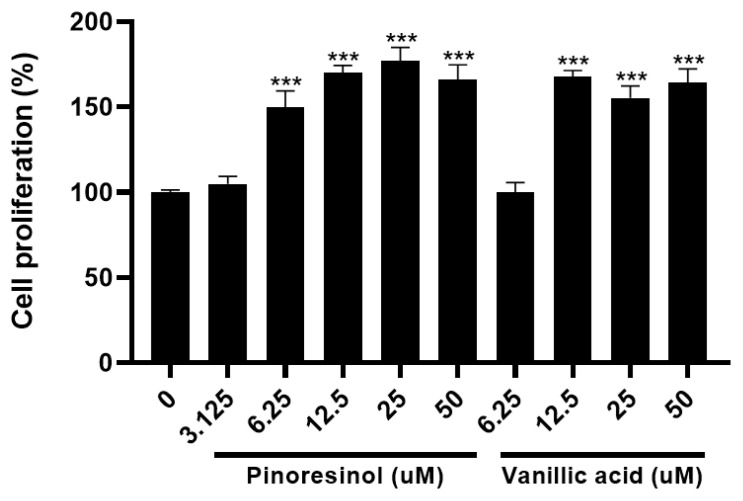
Pinoresinol and vanillic acid facilitating proliferation of mouse myoblasts in a dose-dependent manner. Cell proliferation activity measured by BrdU assay of pinoresinol and vanillic acid at various concentrations on C2C12 cells during the differentiation period. Data represent the mean ± SEM, obtained from at least triplicate determinations. *** indicates *p* < 0.001 calculated by one-way ANOVA.

**Figure 4 molecules-27-05397-f004:**
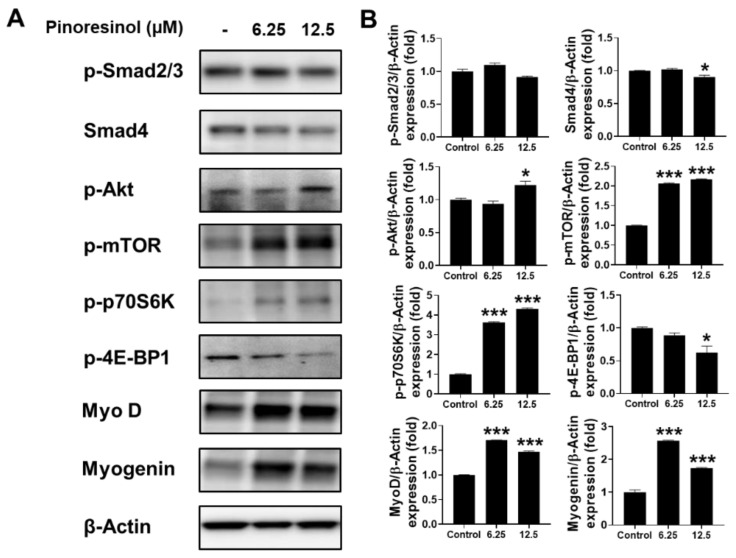
Activated myogenesis by pinoresinol and vanillic acid in mouse myoblast cell line. Western blot using C2C12 cells subjected to different concentrations of pinoresinol (**A**) or vanillic acid (**C**) to assess TGF-β signaling (p-Smad2/3 and Smad 4), IGF-1 signaling (p-Akt, p-mTOR, p-p70S6K, and p-4E-BP1), MyoD, and myogenin. β-Actin was used as the loading control. The relative band intensity of proteins to that of β-actin is expressed as the fold change compared to the control (no treatment) for pinoresinol (**B**) or vanillic acid (**D**). Data are expressed as mean ± SEM, obtained from at least triplicate determinations. * and *** indicate *p* < 0.05 and *p* < 0.001 by one-way ANOVA, respectively.

**Figure 5 molecules-27-05397-f005:**
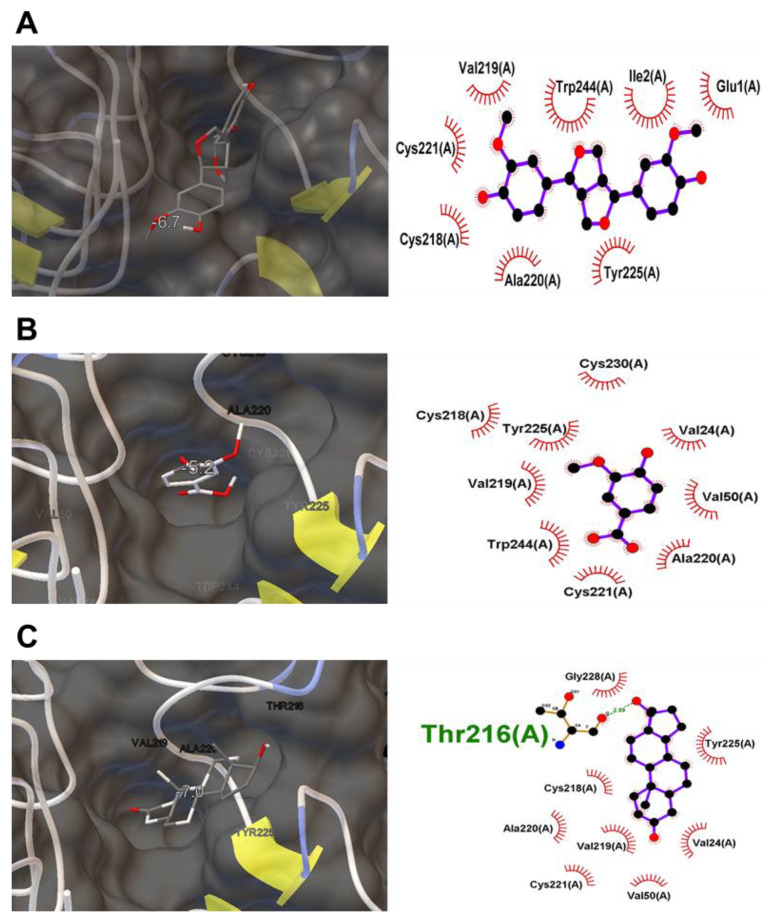
In silico molecular docking analysis for potential small molecules against IGF-1R protein. The left lane represents the interaction of small molecules at the binding site of IGF-1R. The right lane indicates a 2D map of the interactions between small molecules and IGF-1R, including amino acid residue, sequence number and chain. The red lines display hydrophobic interaction, while the green line displays the hydrogen bond. The molecules binding IGF-1R and respective binding energy is as follows: (**A**) Pinoresinol with binding energy −6.7 Kcal/mol, (**B**) Vanillic acid with binding energy −5.2 Kcal/mol, and (**C**) DHT with binding energy −7.0 Kcal/mol.

**Table 1 molecules-27-05397-t001:** Results of docking simulations of two active chemicals (pinoresinol and vanillic acid) with IGF-1R (PDB ID: 1IGR). DHT is dihydrotestosterone used as a positive control.

Receptor	Ligand	Binding Energy (Kcal/mol)
IGF-1R	Pinoresinol	−6.7
	Vanillic acid	−5.2
	DHT	−7.0

## Data Availability

Not applicable.

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
