# Peer review of "Effect of Pinoresinol and Vanillic Acid Isolated from Catalpa bignonioides on Mouse Myoblast Proliferation via the Akt/mTOR Signaling Pathway"

_molecules, 2022, doi:10.3390/molecules27175397_

Round 1

Reviewer 1 Report

The authors describe the isolation of pinoresinol and vanillic acid from Catalpa bignonioides and investigated the effect of these compounds on the Akt/mTOR signaling pathway. The manuscript is well written, and the provided data and the description of the experimental section are sufficient to understand the results of this work. There are not any serious points that require correction except for the following minor points.   

Minor points

1. Add the reference for the differentiation and proliferation assay using C2C12 cells.

2. Add the result of positive control such as steroidal androgen and trimetazine for the cell proliferation assay.

3. It is a little strange that all of the isolated compounds showed approx. 120%. Please check the value of the control. Was the same condition used?

Author Response

  1. Add the reference for the differentiation and proliferation assay using C2C12 cells.

We appreciate the reviewer’s comment and have added references [1-3] for the differentiation and proliferation assay using C2C12 cells to the revised manuscript (line 208).

  1. Add the result of positive control such as steroidal androgen and trimetazine for the cell proliferation assay.

We agree with your suggestion that adding steriodal androgen and trimetazine as positive control in the cell proliferation assay is a way to improve reliability and quality of data. We do not currently have those materials, so they should be placed an order. Unfortunately, because it takes more than 6 weeks to arrive in South Korea, it is impossible to conduct the experiment within the review deadline (10 days).

In a previous paper where a compound extracted from Brown Alga called Ishige Okamurae was treated with C2C12 to measure cell proliferation [4], we used octacosanol as a positive control, included as a reference in the revised manuscript (line 208).

We ask for your understanding of the difficulties in supplying the materials. If the results of positive control are necessary, we will request an extension of the review period and perform the experiment.

  1. It is a little strange that all of the isolated compounds showed approx. 120%. Please check the value of the control. Was the same condition used?

All isolated compounds were tested under the same condition as control (no treatment). The raw data of Fig. 2B is as below.

Control

1

2

3

4

5

6

7

8

10

11

12

13

14

15

97.05882

111.3163

116.1739

161.1025

121.168

151.269

118.8029

104.9941

120.8678

106.7134

113.4449

106.4223

111.1616

107.1864

103.8934

104.0477

153.7961

140.9029

234.6718

131.1101

236.6984

146.3551

131.1479

145.6216

122.2247

141.9465

143.0656

126.467

142.8766

139.1561

98.89348

139.1003

126.5807

170.4074

139.2734

171.5398

140.1227

135.6716

144.4008

125.6055

127.0604

118.5042

131.3542

142.7336

133.9651

References for the reviewers

  1. Wallace, M.A., et al., Overexpression of Striated Muscle Activator of Rho Signaling (STARS) Increases C2C12 Skeletal Muscle Cell Differentiation. Front Physiol, 2016. 7: p. 7.
  2. Wang, L., et al., Sirtuin 1 promotes the proliferation of C2C12 myoblast cells via the myostatin signaling pathway. Mol Med Rep, 2016. 14(2): p. 1309-15.
  3. Girgis, C.M., et al., Vitamin D signaling regulates proliferation, differentiation, and myotube size in C2C12 skeletal muscle cells. Endocrinology, 2014. 155(2): p. 347-57.
  4. Kim, S.Y., et al., Diphlorethohydroxycarmalol (DPHC) Isolated from the Brown Alga Ishige okamurae Acts on Inflammatory Myopathy as an Inhibitory Agent of TNF-alpha. Mar Drugs, 2020. 18(11).

Reviewer 2 Report

In this article, the authors performed the screening of several compoundes isolated from Catalpha bignonioidesfruit extracts to discover effective natural substances for the treatment of muscle loss (sarcopenia).

Briefly, the authors identified 15 compounds and selected two effective compounds (pinoresinol and vanillic acid) based on the ability of mouse myoblast (C2C12 cells) proliferation.

During the process of extraction and isolation, the authors obtained many fractions. It is better to add more clear explanation for the selection of 15 compounds.

Dihydrotestosterone (DHT) has been used as a positive control for Fig. 5 and Table 1. It is better to use DHT in other studies (Fig. 2 to Fig. 4).

In addition to in vitro study, it is better to add in vivo study to demonstrate the impact of pinoresinol and vanillic acid on the treatment of sarcopenia.

Otherwise, the authors need to discuss the impact of pinoresinol and vanillic acid on sarcopenia and/or other human diseases.

For example, the effect of vanillic acid in patients with coenzyme Q10 deficiency has been reported (Biochim Biophys Acta 2014;1824(1):1-6.) and the low level of coenzyme Q10 may be correlated with the incidence of sarcopenia.

Author Response

In this article, the authors performed the screening of several compounds isolated from Catalpha bignonioides fruit extracts to discover effective natural substances for the treatment of muscle loss (sarcopenia).

Briefly, the authors identified 15 compounds and selected two effective compounds (pinoresinol and vanillic acid) based on the ability of mouse myoblast (C2C12 cells) proliferation.

During the process of extraction and isolation, the authors obtained many fractions. It is better to add more clear explanation for the selection of 15 compounds.

The 15 compounds are all isolated ones from methanol extraction of C. bignonioides fruits, not some selected ones. It was confirmed in a previous paper [1] that evaluated α-glucosidase inhibitory activity and insulin secretion effect with the same compounds, which was mentioned in line 64.

Dihydrotestosterone (DHT) has been used as a positive control for Fig. 5 and Table 1. It is better to use DHT in other studies (Fig. 2 to Fig. 4).

We agree with what the reviewer pointed out. As the reviewer suggested, if DHT is used as a positive control in Fig. 2 to Fig. 4, the reliability and quality of the data would be improved. Unfortunately, when any material is ordered from overseas now, it will take about 6 weeks or more to arrive in South Korea for the delay in delivery due to COVID-19, so it is impossible to carry out the suggested experiment within the review deadline (10 days).

We used octacosanol as a positive control in C2C12 cell proliferation assay in a previous paper where a compound extracted from Brown Alga called Ishige Okamurae was treated with C2C12 to measure cell proliferation [2]. We have added this as a reference to the revised manuscript (line 208).

We ask for your understanding of the difficulties in supplying the materials. If the results of positive control are necessary, we will request an extension of the review period and perform the experiment.

In addition to in vitro study, it is better to add in vivo study to demonstrate the impact of pinoresinol and vanillic acid on the treatment of sarcopenia.

Otherwise, the authors need to discuss the impact of pinoresinol and vanillic acid on sarcopenia and/or other human diseases.

For example, the effect of vanillic acid in patients with coenzyme Q10 deficiency has been reported (Biochim Biophys Acta 2014;1824(1):1-6.) and the low level of coenzyme Q10 may be correlated with the incidence of sarcopenia.

We appreciate the reviewer’s valuable comments. It would be better to verify the proliferative effect of pinoresinol and vanillic acid confirmed in C2C12 cells in an in vivo model, which was not included in this study. We would like to perform animal experiments through follow-up studies if there is an opportunity.

We thank for your thoughtful comments with an in vivo literature. Referring to the papers that can suggests the impact of pinoresinol or vanillic acid on sarcopenia and/or other diseases (including the paper given as an example), the following paragraph has been added to the last part of the discussion section in the revised manuscript (line 322):

    Pinoresinol and vanillic acid, demonstrated in this study to promote the proliferation and differentiation of myoblast cells, could be applied to alleviating human diseases such as muscle loss. There have been studies confirming biological effects using those compounds in vivo. Injection of pinoresinol into rats with ischemic damage restored blood flow distribution and reduced the level of reactive oxygen species, whose accumulation is known to be associated with muscle atrophy and muscle weakness [3, 4]. Vanillic acid was reported as a therapeutic option for patients lacking coenzyme Q10 and the lack of coenzyme Q10 is highly correlated with the risk of age-related muscle loss [5, 6]. Although further studies should be done, it is expected that pinoresinol and vanillic acid may be effective in improving muscle strength in humans as well.

References for the reviewer

  1. Oh, Y., et al., The Chemical Constituents from Fruits of Catalpa bignonioides Walt. and Their alpha-Glucosidase Inhibitory Activity and Insulin Secretion Effect. Molecules, 2021. 26(2).
  2. Kim, S.Y., et al., Diphlorethohydroxycarmalol (DPHC) Isolated from the Brown Alga Ishige okamurae Acts on Inflammatory Myopathy as an Inhibitory Agent of TNF-alpha. Mar Drugs, 2020. 18(11).
  3. Lapi, D., et al., Effects of oleuropein and pinoresinol on microvascular damage induced by hypoperfusion and reperfusion in rat pial circulation. Microcirculation, 2015. 22(1): p. 79-90.
  4. Jackson, M.J., Skeletal muscle aging: role of reactive oxygen species. Crit Care Med, 2009. 37(10 Suppl): p. S368-71.
  5. Doimo, M., et al., Effect of vanillic acid on COQ6 mutants identified in patients with coenzyme Q10 deficiency. Biochim Biophys Acta, 2014. 1842(1): p. 1-6.
  6. Fischer, A., et al., Coenzyme Q10 Status as a Determinant of Muscular Strength in Two Independent Cohorts. PLoS One, 2016. 11(12): p. e0167124.

Reviewer 3 Report

The manuscript by Kim et al. reports an interesting study to determine the effects of various metabolites from the Catalpa bignonioides plant on C2C12 myoblast function. The significance in this study primarily lies with identification of natural products that may be leveraged in the future to develop therapeutics to manipulate cell signaling pathways in proliferating myoblasts. However, there are several serious concerns with both the scientific premise and data analysis that render this manuscript unacceptable for publication.

1.    The authors frame the study in the context of studying sarcopenia but focus their study on proliferating C2C12 myoblasts. The use of the C2C12 cell line is not inherently a problem but the premise that impaired myogenesis drives sarcopenia is not well-supported by the literature. It is true that satellite cell/muscle stem cell function declines with age and some studies suggest this cell population may contribute to overall muscle homeostasis. However, the primary model for sarcopenia is that aberrant proteostasis and other cell-intrinsic pathways in mature myofibers drives the decline in muscle mass. Related to this point is that the authors directly state “loss of skeletal muscle mass and function is caused by impaired myogenesis” (introduction, paragraph 1) which is only supported in the context of induced muscle injury. In fact the reference cited for this statement is a review that only refers to factors that contribute to impaired myogenesis and muscle atrophy but does not in any way link the two processes. The statement that “compounds that stimulate skeletal myogenesis and IGF-1 can help overcome age-related skeletal muscle loss” is thus based on flawed or incomplete logic.

2.    The authors show that two compounds, pinoresinol and vanillic acid, have a dose-dependent effect on C2C12 myoblast proliferation using EdU in Figure 2. However, when they moved to immunoblot experiments to determine the effect of these compounds on cell signaling in Figure 3 they used differentiated C2C12 cells. Given that multiple cell signaling pathways fluctuate as C2C12 myoblasts differentiate to myotubes, it is unclear why the experiment was shifted to differentiated C2C12 cells. The authors should perform this blot in proliferating myoblasts. If the authors decide to include blots for phosphorylated signaling molecules in myotubes, they should include a more detailed analysis of the effects of pinoresinol and vanillic acid on differentiation.

3.    In Figure 3, the authors use immunoblotting to measure the levels of several phosphorylated signaling molecules. However, if conclusions about phosphorylation state are to be drawn, the authors must also probe for total amounts of these proteins to ensure any changes are related to activation/phosphorylation rather than levels.

4.    For Figure 3, several of the signaling molecules in question contain multiple phosphorylation sites. The authors must include information about which phosphorylation event is being assayed either in the figure, in the main text, or in the materials and methods section.

5.    The authors conclude that there are significant or non-significant changes in the levels of several phosphorylated proteins in Figure 3. These blots are not quantified and no statistical analysis is included, so the authors cannot state that any of the changes shown are significant or not significant.

6.    The binding prediction shown in Figure 4 is rather difficult to interpret. It is confusing that the positive control substrate, DHT, appears to directly interact with Thr216 (though it is difficult to see in the low-resolution image) but no such interaction occurs with predicted binding of pinoresinol or vanillic acid. The authors should include more explanation of the data and the methodology used to obtain it.

7.    Related to the above point, the authors state in the abstract that both compounds “strongly bound to insulin-like growth factor 1 receptor” but no direct binding assays like isothermal titration calorimetry or fluorescence anisotropy were included. The authors cannot state that the compounds bind IGF-1R based on molecular predictions alone so the language must be changed to accurately reflect the data actually included in the manuscript.

Author Response

Thank you for your insightful and sharp comments on our manuscript. Please see the point-by-point responses in the attachment.

Round 2

Reviewer 2 Report

Please add some references to discuss the impact of DHT on muscle cell proliferation via IGF-1R/Akt/mTOR signaling, in which pinoresinol and vanillic acid may affect.

Author Response

We appreciate the reviewer’s suggestion and have added references [1-3] for DHT on muscle cell proliferation via IGF-1R/Akt/mTOR in the revised manuscript (line 265).

References for the reviewers

  1. Diel, P., et al., C2C12 myoblastoma cell differentiation and proliferation is stimulated by androgens and associated with a modulation of myostatin and Pax7 expression. J Mol Endocrinol, 2008. 40(5): p. 231-41.
  2. Hamzeh, M. and B. Robaire, Androgens activate mitogen-activated protein kinase via epidermal growth factor receptor/insulin-like growth factor 1 receptor in the mouse PC-1 cell line. J Endocrinol, 2011. 209(1): p. 55-64.
  3. Horii, N., et al., Resistance exercise-induced increase in muscle 5alpha-dihydrotestosterone contributes to the activation of muscle Akt/mTOR/p70S6K- and Akt/AS160/GLUT4-signaling pathways in type 2 diabetic rats. FASEB J, 2020. 34(8): p. 11047-11057.

Reviewer 3 Report

While the revised manuscript has addressed many of the concerns raised with the original submission, there are still substantial flaws that render this manuscript unacceptable for publication unless they are addressed.

Authors still suggest in abstract that they showed binding of these compounds to IGF-1R by stating that the compounds “were shown to be stably localized to the active site”. However, this still suggests that the study includes direct experimental evidence of binding. Th authors must explicitly state that molecular docking studies suggest these compounds may bind to active site.

Authors state that proliferation was measured “during the differentiation period” but inducing differentiation causes cell cycle arrest. While there may be some residual proliferation of C2C12 myoblasts prior to cell cycle arrest after transition to low serum medium to induce differentiation, the functional relevance of measuring it as opposed to proliferating myoblasts in normal growth medium is unclear and not commonly included in studies of myoblast function.

The authors’ response includes references to other papers in which phosphorylated signaling molecules are compared to normal loading controls rather than levels of the non-phosphorylated proteins. However, considering that signaling pathways are often regulated by feedback loops that include transcriptional activation/repression and protein stability/turnover it is not acceptable to omit comparison of the phosphoprotein to total protein when drawing conclusions about the phosphorylation state.

Author Response

We would like to thank you and the reviewers for the responses on our manuscript. Please find our point-by-point responses.

----------------------------------------------------------------------

While the revised manuscript has addressed many of the concerns raised with the original submission, there are still substantial flaws that render this manuscript unacceptable for publication unless they are addressed.

Authors still suggest in abstract that they showed binding of these compounds to IGF-1R by stating that the compounds “were shown to be stably localized to the active site”. However, this still suggests that the study includes direct experimental evidence of binding. Th authors must explicitly state that molecular docking studies suggest these compounds may bind to active site.

We thank for the reviewer’s accurate comment. We have changed the sentence in the abstract to “the results of in silico molecular docking study showed that they may bind to the active site” in the revised manuscript.

Authors state that proliferation was measured “during the differentiation period” but inducing differentiation causes cell cycle arrest. While there may be some residual proliferation of C2C12 myoblasts prior to cell cycle arrest after transition to low serum medium to induce differentiation, the functional relevance of measuring it as opposed to proliferating myoblasts in normal growth medium is unclear and not commonly included in studies of myoblast function.

We appreciate the reviewer’s in-depth comment. It is well known that when C2C12 cells are exposed to a low serum medium for differentiation, they have fused to form multinucleated myotubes and no longer divide. But improvement in skeletal muscle function is affected by both an increase in the muscle cell number by division and elongation due to fusion during differentiation, so we thought that it would be appropriate to examine both [1, 2] and that it is considered a suitable way to evaluate proliferation during differentiation process.

As the reviewer mentioned, brief division before cell cycle arrest may not be a convincing explanation. However, previous papers from not only ours [3, 4] but also other research group [5] that have evaluated the efficacy of natural products in skeletal muscle proliferation have examined cell proliferation in differentiated skeletal muscle cells.

To address the reviewer’s concern, the following paragraph has been added to the discussion section in the revised manuscript (line 322):

    The proliferative effect of both compounds was evaluated during the differentiation process of C2C12, known that cell cycle is arrested. However, since the function of skeletal muscle is represented by differentiation proceeding with an increase in the number of muscle cells, we conducted measuring cell proliferation in differentiated muscle cells as well established in previously published papers [3-5].

We expect the reviewer to understand our decision.

The authors’ response includes references to other papers in which phosphorylated signaling molecules are compared to normal loading controls rather than levels of the non-phosphorylated proteins. However, considering that signaling pathways are often regulated by feedback loops that include transcriptional activation/repression and protein stability/turnover it is not acceptable to omit comparison of the phosphoprotein to total protein when drawing conclusions about the phosphorylation state.

We appreciated the reviewer’s comment. As the reviewer pointed out, signaling pathways may be regulated by feedback loops such as transcriptional activation/repression and protein stability/turnover thereby the amount of total protein should be determined, which is a limitation in our study.

To resolve this, it is necessary to purchase antibodies against Smad2/3, Akt, mTOR, p70S6K, and 4E-BP1. Unfortunately, when any material is ordered from overseas now, it will take about 6 weeks or more to arrive in South Korea for the delay in delivery due to COVID-19, so it is impossible to carry out the suggested experiment within the review deadline (10 days).

In addition to the references from MDPI as mentioned in the previous answer [6-8], several recent papers published in leading journals [9-13] also compared the level of protein phosphorylation with β-actin as control.

We are well aware that references may not provide a complete answer to the reviewer’s comment, but we ask for the reviewer’s thoughtful understanding of the difficulties in supplying the materials. If the results of the total proteins are necessary, we will request an extension of the review period and perform the experiment.

References for the reviewers

  1. Diel, P., et al., C2C12 myoblastoma cell differentiation and proliferation is stimulated by androgens and associated with a modulation of myostatin and Pax7 expression. J Mol Endocrinol, 2008. 40(5): p. 231-41.
  2. Perie, L., et al., Enhancement of C2C12 myoblast proliferation and differentiation by GASP-2, a myostatin inhibitor. Biochem Biophys Rep, 2016. 6: p. 39-46.
  3. Oh, M., et al., Phytochemicals in Chinese Chive (Allium tuberosum) Induce the Skeletal Muscle Cell Proliferation via PI3K/Akt/mTOR and Smad Pathways in C2C12 Cells. Int J Mol Sci, 2021. 22(5).
  4. Kim, S.Y., et al., The Effects of Marine Algal Polyphenols, Phlorotannins, on Skeletal Muscle Growth in C2C12 Muscle Cells via Smad and IGF-1 Signaling Pathways. Mar Drugs, 2021. 19(5).
  5. Kalhotra, P., et al., Phytochemicals in Garlic Extract Inhibit Therapeutic Enzyme DPP-4 and Induce Skeletal Muscle Cell Proliferation: A Possible Mechanism of Action to Benefit the Treatment of Diabetes Mellitus. Biomolecules, 2020. 10(2).
  6. Gu, H., et al., Bee Venom and Its Major Component Melittin Attenuated Cutibacterium acnes- and IGF-1-Induced Acne Vulgaris via Inactivation of Akt/mTOR/SREBP Signaling Pathway. Int J Mol Sci, 2022. 23(6).
  7. Gassowska-Dobrowolska, M., et al., Alterations in Tau Protein Level and Phosphorylation State in the Brain of the Autistic-Like Rats Induced by Prenatal Exposure to Valproic Acid. Int J Mol Sci, 2021. 22(6).
  8. Conte, M., et al., Gliadin Peptide P31-43 Induces mTOR/NFkbeta Activation and Reduces Autophagy: The Role of Lactobacillus paracasei CBA L74 Postbiotc. Int J Mol Sci, 2022. 23(7).
  9. Yang, Y., et al., Natural pyrethrins induce autophagy of HepG2 cells through the activation of AMPK/mTOR pathway. Environ Pollut, 2018. 241: p. 1091-1097.
  10. Ji, B., et al., Activation of the P38/CREB/MMP13 axis is associated with osteoarthritis. Drug Des Devel Ther, 2019. 13: p. 2195-2204.
  11. Conza, D., et al., Metformin Dysregulates the Unfolded Protein Response and the WNT/beta-Catenin Pathway in Endometrial Cancer Cells through an AMPK-Independent Mechanism. Cells, 2021. 10(5).
  12. Cheng, B.F., et al., Neural Cell Adhesion Molecule Regulates Osteoblastic Differentiation Through Wnt/beta-Catenin and PI3K-Akt Signaling Pathways in MC3T3-E1 Cells. Front Endocrinol (Lausanne), 2021. 12: p. 657953.
  13. Pu, Y., et al., The immunomodulatory effect of Poria cocos polysaccharides is mediated by the Ca(2+)/PKC/p38/NF-kappaB signaling pathway in macrophages. Int Immunopharmacol, 2019. 72: p. 252-257.

Round 3

Reviewer 3 Report

The authors have adequately addressed all concerns.